# Valorization of Cereal Byproducts with Supercritical Technology: The Case of Corn

**Ádina L. Santana** [1,2,*] and **Maria Angela A. Meireles** [2,*]

1 Grain Science and Industry Department, Kansas State University, 1301 N Mid Campus Drive, Manhattan, KS 66506, USA
2 School of Food Engineering, University of Campinas (UNICAMP), R. Monteiro Lobato 80, Campinas 13083-862, SP, Brazil
* Correspondence: adina.santana@gmail.com or adina@ksu.edu (Á.L.S.); maameireles@lasefi.com (M.A.A.M.)

**Abstract:** Ethanol and starch are the main products generated after the processing of corn via dry grinding and wet milling, respectively. Milling generates byproducts including stover, condensed distillers' solubles, gluten meal, and the dried distillers' grains with solubles (DDGS), which are sources of valuable compounds for industry including lignin, oil, protein, carotenoids, and phenolic compounds. This manuscript reviews the current research scenario on the valorization of corn milling byproducts with supercritical technology, as well as the processing strategies and the challenges of reaching economic feasibility. The main products recently studied were biodiesel, biogas, microcapsules, and extracts of enriched nutrients. The pretreatment of solid byproducts for further hydrolysis to produce sugar oligomers and bioactive peptides is another recent strategy offered by supercritical technology to process corn milling byproducts. The patents invented to transform corn milling byproducts include oil fractionation, extraction of undesirable flavors, and synthesis of structured lipids and fermentable sugars. Process intensification via the integration of milling with equipment that operates with supercritical fluids was suggested to reduce processing costs and to generate novel products.

**Keywords:** yellow corn; biowaste; supercritical $CO_2$; biorefinery; process intensification; cost of manufacture

## 1. Introduction

Corn (*Zea mays* L.), also known as yellow or sweet corn, is a multipurpose crop used for many industrial applications including foods, fuels, cosmetics, and pharmaceuticals.

According to a recent report published by the United States Department of Agriculture, the three main producers of corn worldwide are the United States (353.84 million ton/year), China (274 million ton/year), and Brazil (126 million ton/year) [1].

Corn processing provides opportunities to generate multiple products including flour, ethanol, starch, protein, and syrups. Before processing, the grains are separated from the stover (leaves, stalks, cobs, and silk), which is an underutilized mixture of lignocellulosic material. The stover is produced at an equal proportion of grains processed, i.e., 1 kg of corn grain generates 1 kg of stover [2,3].

The mill is the main form of processing corn. The processes for milling corn include (a) dry milling, (b) dry grinding, and (c) wet milling.

According to Stanford and Keener [4], the term dry milling is erroneously used throughout the literature to describe both dry milling and dry grinding processes.

Dry milling consists of mechanical procedures to separate the germ, tip cap, and pericarp from the endosperm to produce flour, oil, grits, and corn meal. These are products commonly sold for human consumption and hominy feed, which is a byproduct sold for animal feeding [4,5].

Dry grinding is destined to produce ethanol via the fermentation of corn kernels. Wet milling differs from dry grinding in the utilization of huge amounts of water to fractionate corn kernels into starch, oil, protein, fiber, ethanol, and sweeteners [6,7].

The corn milling industry plays an important role in the corn supply chain. It starts from crop production and ends with feed and food industrial channels including retail markets, animal feed, food processors, cosmetics, and fuel, as well as the food supply for emergency government food aids [5]. In the United States, some companies that mill corn include Cargill [8], Bunge [9], Gavilon [10], and J-Six Enterprises [11].

The generation of underutilized fractions after crop processing is unavoidable. In addition, auto-oxidation of lipids, microbial growth in high moisture materials, and changes in enzymatic activity accelerates the decomposition of discarded fractions, and such difficulty in the reutilization of products turns into an environmental concern [12].

The undervalued fractions from dry grinding include the condensed distillers, and the wet or dried distiller's grains with solubles, whereas the byproducts derived from wet milling include spent germ meal, gluten feed, oil, and steep water. The stover is an undervalued fraction generated before corn grain processing. Even with the known utilization of some byproducts in animal feed [7,13,14], dry grinding and wet milling face a challenge in the valorization of byproducts.

Supercritical technology includes multiple separation methods that use solvents operating in the region of compressed liquid, and subcritical and supercritical state in which carbon dioxide ($CO_2$) is the most preferred solvent. Supercritical $CO_2$ (SC-$CO_2$) is a low-cost, non-flammable, and non-toxic solvent that shows high efficiency in extracting and fractionating low-polar products, including oils and waxes [15–17], to dry particles [18] and assist in chemical reactions [19]. As reviewed by Fărcaș and coworkers [20], SC-$CO_2$ provokes low oxidative and thermal impact in the products generated after the processing of undervalued cereal products.

In this context, supercritical technology emerged as a green approach to improve the processing of crops.

This review discusses the current opportunities offered by the application of supercritical technology to transform byproducts derived from dry grinding and wet milling into value-added products, including oil [21], wax [17], biofuels [22], biogas [23], and microparticles [18]. The patents survey and our opinions on the processing strategies to reduce costs and improve the quality of products were also included.

## 2. Dry Grinding

Corn is abundantly processed into ethanol via dry grinding because of the high conversion rates of corn starch.

Approximately 450 million bushels of corn are used for both dry grinding and wet milling to produce ethanol [24]. One bushel of corn is equivalent to 25.40 kg. Before grain processing, there is a generation of stover, a fraction consisting of leaves, cob, straw, and silk [25].

After harvesting corn, the stover is separated from the grains, and the grains are cleaned, stored, and tempered. The tempering increases the moisture of corn kernels to facilitate the degerming process, i.e., the separation of germ and bran coat [5].

The grains are ground to a fine powder, known as meal, which is mixed with water and heated at 85 °C. Afterward, the enzyme alpha-amylase is added, followed by heating at 110–150 °C for 1 h, to accelerate the conversion of starch into dextrose, resulting in a liquefied slurry (Figure 1).

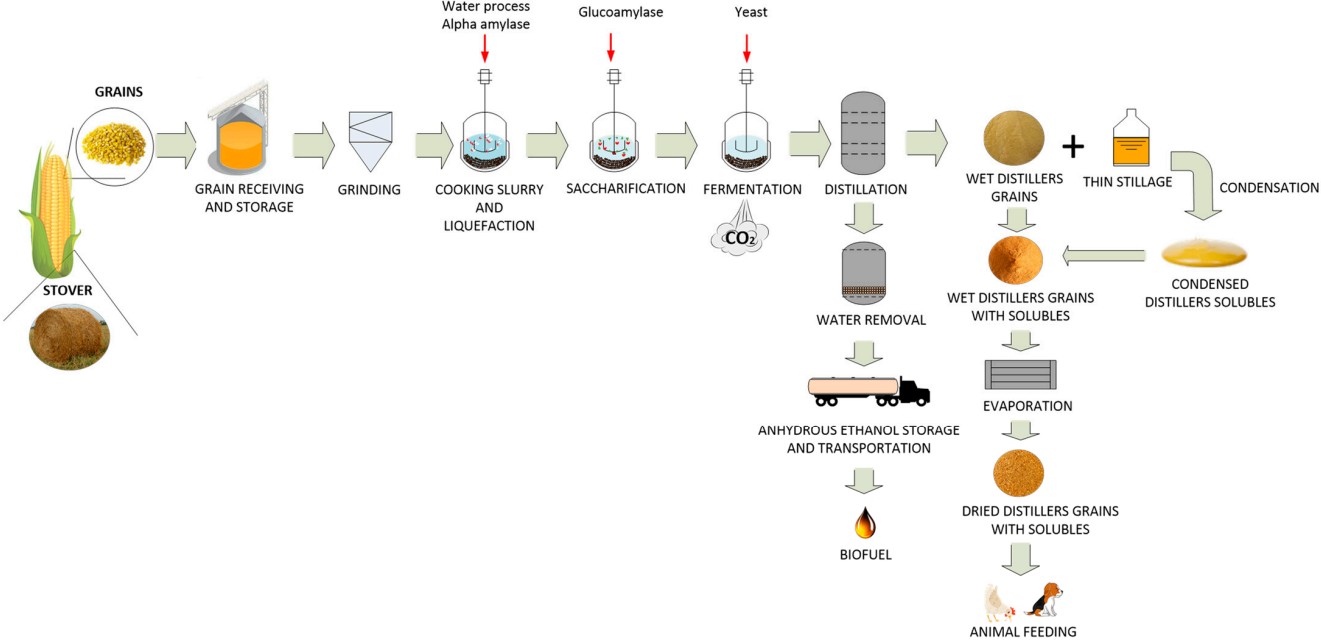

**Figure 1.** The steps involved in corn dry grinding.

The slurry is cooled to room temperature. Afterward, glucoamylase is added to saccharify the dextrose in the slurry into glucose. Glucose is the substrate for fermentation by yeast strains (*Saccharomyces cerevisiae*). The glucose fermentation results in two streams, one consisting of $CO_2$, and the other consisting of beer which is a slurry composed of up to 16% ethanol by volume [26].

The beer is pumped into distillation columns resulting in wet ethanol and the stillage. Wet ethanol is distilled up to 95.6% azeotropic point, and then transferred to the molecular sieve system to produce anhydrous ethanol. The stillage is a viscous liquid composed of 6–16% solids.

The stillage is dried, resulting in wet distillers' grains and thin stillage (TS). The storage and handling of wet distillers' grains (WDG) increase processing costs because they can be stored for up to 5–7 days and require the use of chemicals to inhibit microbial growth. In addition, the water in TS is evaporated resulting in the condensed distillers' solubles (CDS), a highly viscous syrup.

The drying of CDS demands high energy because of high syrup viscosity. Thus, to reduce the quantity of streams, one proposed solution was to mix the CDS with WDG and, subsequently, drying of the resulting mixture on the dried distillers' grains with solubles (DDGS), which has longer shelf life due to its low moisture (10–12%). Another possibility is to use the TS or the CDS in animal diets [27]. By the end of dry grinding, one bushel of corn can produce approximately 10.2 L ethanol and 8 kg DDGS [28,29].

DDGS is donated or sold to be used as an animal diet component or as fertilizer. The disadvantage of using DDGS as fertilizer is the excess of organic matter leading to soil pollution. In addition, there is a loss of nutrients from corn DDGS that could be reused. In 2019, dry grinding generated 22,591,477 tons of DDGS [30].

## 3. Wet Milling

The wet milling process separates the corn kernels into starch, oil, protein, ethanol, and fiber. Grains are received, separated from foreign materials, and stored. Afterward, the grains are inserted into a stainless-steel tank of up to 600 metric tons for steeping, mixed with an aqueous solution consisting of 0.12–0.20% sulfur dioxide at 50 °C for 24–48 h to prevent microbial growth and to get the germ easily separated from the endosperm.

Lactic acid produced by *Lactobacillus* bacteria in the steeping water induces the kernel's softening and enhances sulfur dioxide's sorption. Sulfur dioxide prevents putrefaction and helps to separate the starch from the protein because it acts as a reducing agent in breaking disulfide bonds within the protein, enhancing starch recovery [31].

The steeping process generates a stream consisting of steepwater which is a nutritive media composed of 5–10% solids including sugars and protein [6,32].

The steeped corn is ground and subsequently transformed via multiple separation steps. The oil is extracted from the germ via solvent extraction generating a byproduct known as germ meal (Figure 2). The mixture consisting of fiber, starch, and protein is inserted into hydrocyclones and washing screens to separate the fiber from the starch and protein (millstarch). The fiber is dried and pressed resulting in the corn gluten feed.

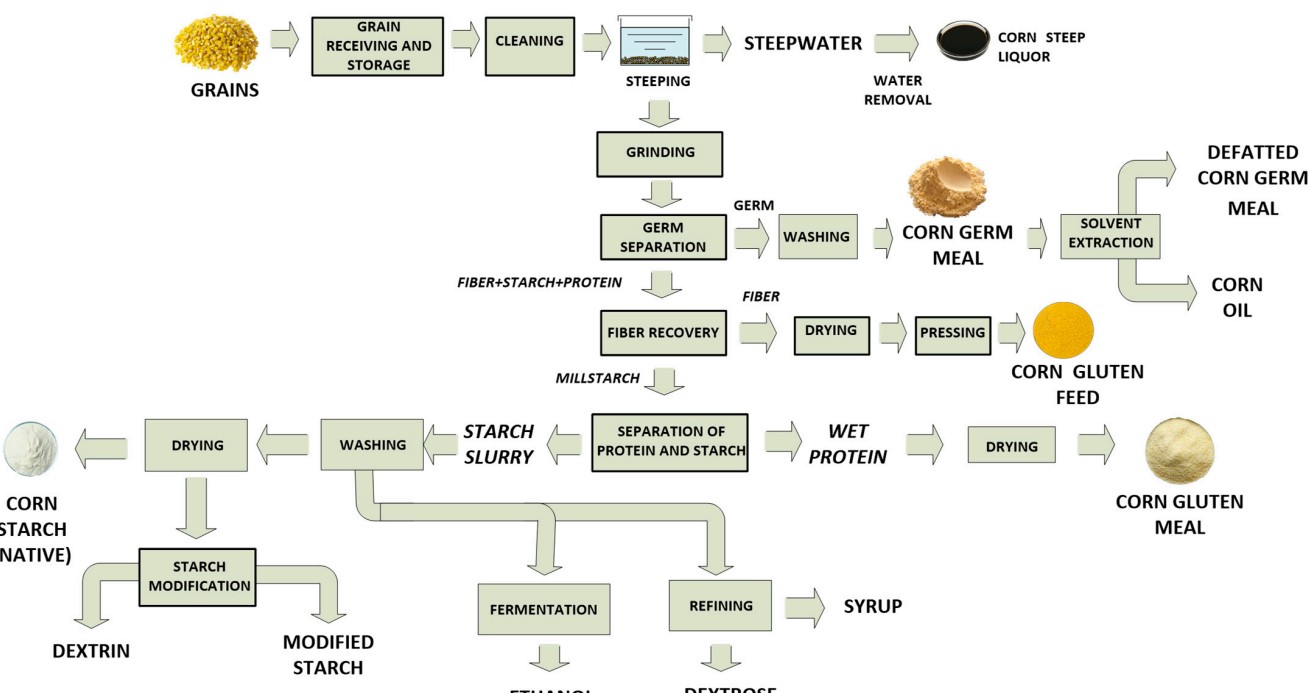

**Figure 2.** The steps involved in corn wet milling.

The millstarch is centrifuged to separate the starch and gluten meal. The gluten meal is dried, whereas the starch is washed several times to recover the granules by reducing soluble impurities and then, subsequently dried. The starch can be sold as such or used as a raw ingredient to produce ethanol via fermentation or to produce sweeteners via refining.

At the end of the wet milling process, one bushel of corn produces 6–9 L ethanol, 14 kg starch, 0.5–0.9 kg oil, 5–6.4 kg corn gluten feed, and 0.9–1.4 kg corn gluten meal [4]. In 2019, wet milling produced approximately 765,620 tons of corn germ meal, 3,468,292 tons of corn gluten feed, and 106,899 tons of gluten meal [30]. In addition, it is difficult in wet milling to reduce the water used for processing. For instance, 1.5–1.78 L of water processes 1 kg of corn. In addition, the amount of wastewater generated is equivalent to a medium-large city.

Furthermore, another concern faced by wet milling includes the environmental damage provoked by air pollution. Air pollution may be provoked by acid rain via sulfur dioxide emission from the steeping step, or particulate material and smoke from evaporators during drying. Aside from obtaining different products, wet milling demands high capital and operational costs. For instance, the cost of investment to construct a plant that processes 2542 tons of grains daily was estimated at USD 200–300 million [6].

## 4. Dry Grinding and Wet Milling Byproducts

### 4.1. Condensed Distillers' Solubles (CDS)

The condensed distillers' solubles (CDS) consist of a blend of corn and yeast after the fermentation step in dry grinding. Enzymatic hydrolysis transforms CDS protein into small peptides and amino acids with antioxidant potential against free radicals [33], and the potential to inhibit angiotensin I-converting enzyme (ACE) which is a protein that regulates the volume of fluids in the body thereby allowing an increase in blood pressure [34].

### 4.2. Dried Distillers' Grains with Solubles (DDGS)

The dried distiller's grains with solubles (DDGS) are a low-cost source of nutrients, including lipids, carotenoids, and protein. The nutrient composition supports studies on DDGS use in poultry, swine, and beef cattle diets [35].

According to Heuzé and coworkers [36], 63% of distillers' grains derived from corn are sold as dried products. Recently, Langemeier [37] estimated the average price of corn DDGS at USD 165/ton.

Xanthophylls are oxygenated carotenoids in corn DDGS that play a role in visual and cognitive development. Shin and coworkers [14] observed that lutein was the predominant xanthophyll found in corn DDGS, with a concentration of 4.4 to 9.3 times higher than zeaxanthin. In addition, the same authors detected tocopherols and tocotrienol, where gamma-tocopherol and gamma-tocotrienol were predominant in multiple corn DDGS samples.

Lutein is commercially extracted from marigold petals, costing USD 34.33/lb [38]. In this case, corn DDGS could be reused to decrease the costs of lutein.

Phenolic compounds were detected in corn DDGS, including vanillic, caffeic, p-coumaric, ferulic, and sinapic acids. The ferulic acid concentration detected in the DDGS was around three-fold higher than in crude corn [39]. Phenolic compounds have been extensively studied for their health benefits against cancer [40] and other illnesses.

In addition, corn DDGS contains low-cost oil, also known as distillers corn oil, from which antioxidants, tocopherols, and tocotrienols have been detected [14]. Based on this, some ethanol producers took the opportunity to extract oil mechanically from the WDG before drying and selling it as a secondary product. However, the semi-defatted corn DDGS distributed to animal feeding provoked problems attributed to the reduced energy value in the meal by 45 kcal/lb per % oil extracted [35,41].

### 4.3. Germ Meal

Germ meal is the low-fat solid fraction obtained after the hexane extraction of corn in wet milling. Germ meal is a source of oleic- and linoleic-fatty acids [21], carotenoids, and protein. Corn germ meal was studied mostly for animal feeding [7,13]. Zeaxanthin (989 µg/kg) and lutein (72 µg/kg) were also found in corn germ [42]. Recently, germ meal has been shown to be a source of protein with technological properties desirable for the industry [43].

### 4.4. Gluten Feed and Gluten Meal

Gluten feed is obtained after the separation of fiber from protein and starch. The ingestion of PROMITOR®, a commercial soluble fiber derived from corn, was studied by Whisner and coworkers [44], who observed the enhanced quality in the gut microbiome of female adolescents, allowing the increase of calcium absorption. In addition, Costabile and coworkers [45] showed that the ingestion of PROMITOR® with the probiotic *Lactobacillus rhamnosus* resulted in a positive synergy by decreasing inflammation, and by inhibiting the activity of interleukin 6 which is a proinflammatory cytokine [46].

Gluten meal is the protein byproduct obtained after separation from starch. Zein and glutelin are proteins of high amino acid content obtained from gluten meals. Zein is an alcohol-soluble product with 45–50% of protein which is not used directly for human consumption due to its hydrophobicity. The hydrophobic nature of zein redirected its application for the formulation of films and coatings [47]. Information on the utilization of corn glutelin in foods is scarce.

Peptides converted from corn gluten meal could in vitro scavenge free reactive oxygen species by increasing the levels of antioxidant enzymes [48,49].

### 4.5. Stover, Silk, and Steep Liquor

Corn stover, or straw, consists of plant parts left behind after grain harvesting including leaves, stalks, cobs, and silk. Corn stover is a lignocellulosic feedstock for energy production including polyol fuel via pyrolysis [50] and ethanol via enzymatic hydrolysis of pretreated stover [51]. Corn cob found a recent application as an activated carbon adsorbent for mercury removal [52].

Corn silk, also referred to as corn hair or Maydis stigma, is a byproduct of corn stigma and style used in Chinese medicine to treat multiple diseases including inflammation in the urinary tract [53]. The rich composition of phenolic compounds is strongly linked to corn silk's action against inflammation including maysin and caffeoylquinic acid derivatives [54]. Isolated maysin from corn silk showed in vitro effects against obesity by inhibiting the functionality of adipocytes [55].

Steep liquor is the waste generated after steeping in the wet milling process. Rodriguez-Lopez et al. [56] detected phenolic compounds and protein in corn steep liquor, which may serve as food nutrients. In addition, steep liquor was studied as a nutrient feedstock for fermentation [57].

Since it was administered at low dosages, corn's nutrients may contribute to agriculture as a growth stimulant and fertilizer [58].

The reviewed proximate composition and the bioactive compounds detected in corn milling byproducts are shown in Tables 1 and 2, respectively.

**Table 1.** The proximate composition of corn milling byproducts (%).

| | Moisture | Protein | Fat | Ash | Carbohydrates | | | | | | | Reference |
|---|---|---|---|---|---|---|---|---|---|---|---|---|
| | | | | | Total * | Fiber | Starch | Glucan | Cellulose | Lignin | Xylan | |
| DDGS | 10 | 27.5 | 10.5 | 4.1 | 47.9 | 32.8 | 5.2 | 21.2 | 16 | - | 8.2 | [59,60] |
| TS | 91.78 | 0.12 | 1.75 | 0.74 | 5.61 | 1.27 | - | - | - | - | - | [61] |
| CDS | 68.51 | 8.11 | 1.86 | 4.17 | 17.35 | 0.32 | - | - | - | - | - | [62] |
| Stover | 11.8 | 9.01 | 0 | 0 | 79.19 | 17.9 | - | 31.7 | 32.9–43.1 | 5.4–12.6–14.8 | 17.1 | [17,63–65] |
| Gluten feed | 10 | 20.3 | 3 | 7.4 | 59.3 | 32 | 15.5 | - | - | - | - | [59] |
| Gluten meal | 10 | 28.37 | 2.2 | 5.83 | 53.6 | 5.88–7.3 | 13.9 | | 13.64 | | | [59,66] |
| Germ meal | 10 | 25.12 | 2.73 | 3.8 | 58.35 | 35.5 | 17.8 | - | - | - | - | [13,59] |
| Steep liquor | 50 | 16.4 | 2.3 | 1.5 | 29.8 | 1.8 | 5.7–14.8 | - | - | - | - | [7,59,67] |

* Total carbohydrates estimated by subtraction.

**Table 2.** The recently reviewed bioactive compounds identified in corn milling byproducts.

| Substance | DDGS | Germ Meal | Gluten Feed Fiber | Gluten Meal | Thin Stillage | Stover | Reference |
|---|---|---|---|---|---|---|---|
| Carotenoids | Lutein: 101 µg/g | Lutein: 0.072 µg/g | Lutein: 5.24 µg/g | Lutein: 2.5–15 µg/g | Lutein: 31 µg/g Zeaxanthin: 32.5 µg/g | - | [42,61,68–70] |
| | Zeaxanthin: 69 µg/g | Zeaxanthin: 0.989 µg/g | Zeaxanthin: 6.61 µg/g | Zeaxanthin: 5–35 µg/g | Beta cryptoxanthin: 0.70 µg/g | - | |
| | Beta cryptoxanthin: 47 µg/g | Beta carotene: 0.16 µg/g | Beta-carotene:0.299 µg/g | Beta-carotene:1.22 µg/g | - | - | |
| | - | - | - | - | - | - | |
| Tocopherols | Alpha-tocopherol: 0.108 µg/g | Alpha-tocopherol: 31.74 µg/g | Alpha-tocopherol: 0.28 µg/g | Alpha-tocopherol: 0.18 µg/g | Alpha-tocopherol: 37.3 µg/g | - | [14,69] |
| | Gamma tocopherol: 0.069 µg/g | Gamma tocopherol: 535.59 µg/g | Gamma tocopherol: 9.78 µg/g | Gamma tocopherol: 13.96 µg/g | Gamma tocopherol: 143 µg/g | - | |
| | Delta tocopherol: 0.0182 µg/g | Delta tocopherol:17.38 µg/g | Delta tocopherol: 1.08 µg/g | Delta tocopherol: 0.97 µg/g | Delta tocopherol: 4.46 µg/g | - | |
| Tocotrienols | Alpha Tocotrienol: 0.0093 µg/g | Alpha Tocotrienol: 0.32 µg/g | Alpha Tocotrienol: 1.32 µg/g | Alpha Tocotrienol: Non identified | Alpha Tocotrienol: 29.6 µg/g | - | [14,69] |
| | Gamma tocotrienol: 0.014 µg/g | Gamma tocotrienol: 3.67 µg/g | Gamma tocotrienol: 6.61 µg/g | Gamma tocotrienol: 4.52 µg/g | Gamma tocotrienol: 38.5 µg/g | - | |
| | | Delta tocotrienol: 0.24 µg/g | Delta tocotrienol: 0.54 µg/g | Delta tocotrienol: 0.50 µg/g | Delta tocotrienol 0.09 µg/g | - | |
| Phenolic compounds | Ferulic acid: 7010 µg/g | Ferulic acid: 9870—636,540 µg/g | Ferulic acid: 1020–4200 µg/g | - | - | Ferulic acid: 990.20–3515.50 µg/g- | [39,71–73] |
| | p-Coumaric acid: 530 µg/g | - | p-Coumaric acid: 60–340 µg/g | - | - | p-Coumaric acid: 2027.45–7307.75 µg/g | |
| | Caffeic acid: 770 µg/g | - | - | - | - | - | |
| | Vanillic acid: 500 µg/g | - | - | - | - | - | |
| | Sinapic acid: 8780 µg/g | - | - | - | - | - | |
| Phytosterols | Campesterol: 2916 µg/g | Campesterol: 87 µg/g | Campesterol: 201.1 µg/g | Campesterol: 87.2 µg/g | Beta sitosterol: 1467 µg/g | Campesterol: 226.4 µg/g | [17,61,68,69] |
| | Stigmasterol: 861 µg/g | Stigmasterol: 343.8 µg/g | Stigmasterol: 476.1 µg/g | Beta sitosterol: 1342.5 µg/g | Squalene: 168 µg/g | Stigmasterol: 319.6 µg/g | |
| | Sitosterol: 9066 µg/g | Beta sitosterol: 311.1 µg/g | Beta sitosterol: 2943.6 µg/g | - | Δ-5 avenasterol: 455 µg/g | Beta-sitosterol: 735.6 µg/g | |
| | Sitostanol: 4186 µg/g | - | - | - | - | - | |

## 5. Supercritical Technology

Supercritical technology is a term that includes a wide range of separation methods that operate at the phase regions of compressed liquid, and subcritical and supercritical fluid. Biofuels, biopolymers, and extracts rich in antioxidants are among the value-added products generated via supercritical technology as opportunities to valorize corn fractions, reduce costs, and promote sustainability for the process.

The reusability, low cost, low toxicity, non-flammability, and moderate critical point (31.2 °C and 7.38 MPa) allow the use of $CO_2$ as the preferred supercritical fluid. In addition, SC-$CO_2$ is a carrier solvent that does not react with the extract, thereby avoiding contamination [15].

The components in the supercritical technology-based apparatus include a $CO_2$ storage tank, solvent reservoir, pump, heat and cooling devices, temperature and pressure indicators, valve assembly, flowmeter and high-pressure vessel or chamber (Figure 3(A.1)). Figure 3 shows a scheme of some apparatus for some techniques that use supercritical fluids. The configurations are modified according to the research method and goals.

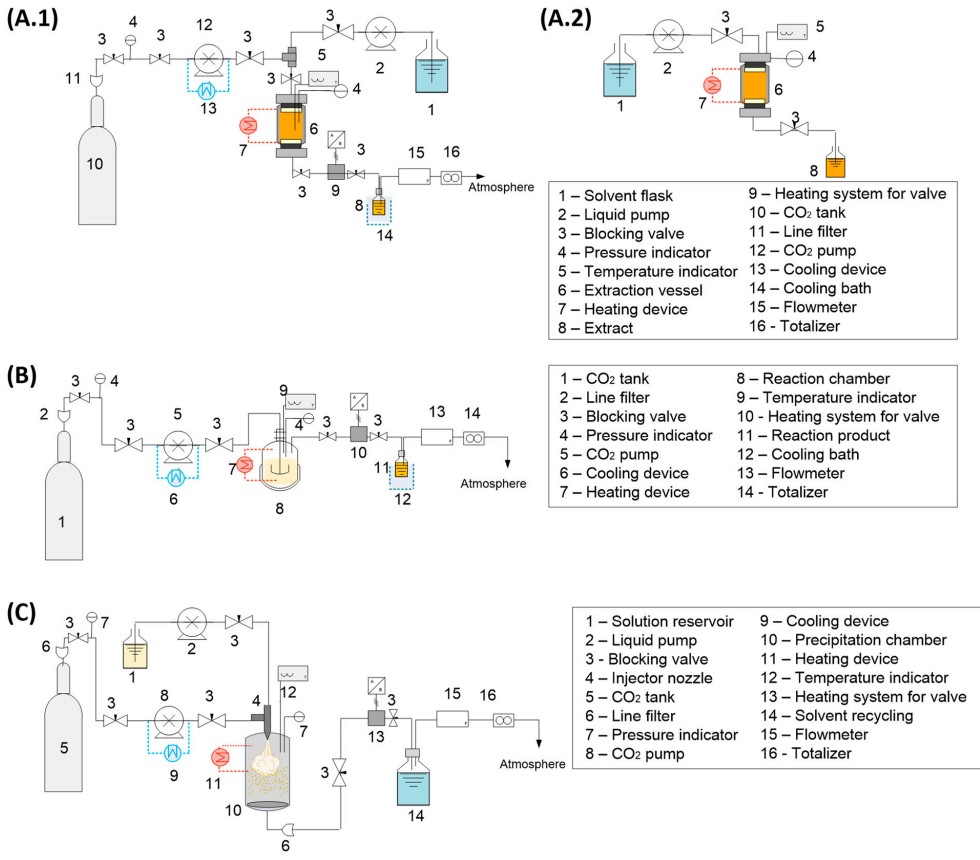

**Figure 3.** The schematic drawing of equipment designed for SFE with cosolvent (**A.1**), PLE (**A.2**), the chemical reaction in a batch reactor (**B**) and particle formation (**C**).

Details on costs and procedures to construct supercritical technology-based equipment are available elsewhere [74,75].

### 5.1. Processes

5.1.1. Supercritical Fluid Extraction (SFE) and Hydrolysis Assisted with SC-$CO_2$

Supercritical fluid extraction (SFE) has been used mostly to extract non-polar products including oils [21] and waxes [17] from corn milling byproducts.

Marinho and coworkers [21] observed that the SFE of oil from corn germ (45–85 °C, 15–25 MPa) was improved by including ethanol as a cosolvent in terms of the antioxidant

potential detected in extracts. The fatty acids oleic and linoleic were predominant in the extracts.

Burlini and coworkers used SC-CO$_2$ to extract phytosterols from corn germ [71]. Previous studies used sequential extractions to obtain different products from crops which may be useful for further studies with corn. For instance, studies with turmeric optimized the extraction of essential oils with SC-CO$_2$, followed by the extraction of phenolic compounds and monosaccharides with pressurized solvents [76–78].

Espinosa-Pardo [43] and coworkers performed the extraction of corn germ by first extracting oil with SC-CO$_2$ (45 °C, 40 MPa, 5.6 g/min, without co-solvents) as a pretreatment strategy to prepare the raw material for protein extraction. As the authors mentioned, SFE provided a semi-defatted corn germ meal and an extract rich in phenolic compounds. In addition, the protein extracts obtained from semi-defatted corn germ meal possessed industry-interest technological properties, including foaming properties higher than 80%.

Even with proven applications in human health, there are no recent reports on the processing of corn silk with supercritical technology. Liu and coworkers [79] studied the SFE of flavonoids from corn silk using the Box-Behnken design combined with Response Surface Methodology. The authors did a sequential extraction, using pure carbon dioxide for 2 h, followed by extraction with ethanol [79].

Recently, Sun and coworkers [80] used SC-CO$_2$-assisted hydrolysis of corn cob and stalk as a pretreatment strategy to improve the release of oligosaccharides for further enzymatic hydrolysis. The authors hypothesized that the pretreatment with SC-CO$_2$ improved the utilization of lignocellulosic sugars. This behavior can be explained by the generation of carbonic acid due to the interaction between CO$_2$ and the water present in the raw material causing temporary acidification of the system [80].

In another study, supercritical CO$_2$ (120–170 °C) combined with ultrasound pretreated corn cob and stalk improved the sugar yield by 75% after enzymatic hydrolysis of pretreated raw material [81].

### 5.1.2. Subcritical Water-Assisted Hydrolysis

Water at subcritical conditions (100–374 °C, pressures up to 22.1 MPa) has been used as a green solvent to valorize lignocellulosic materials via bioactive compounds extraction or via the generation of fermentable sugars by hydrolysis [82]. The process using pressurized water at temperatures lower than 100 °C is known as pressurized hot water extraction [15]. Subcritical water is recommended for extraction in pseudo-continuous mode, which is performed via the collection of extracts after water flows across the extraction vessel (Figure 3(A.2)).

For the pretreatment of lignocellulosic materials via hydrolysis, subcritical water is injected in batch mode, i.e., the wet raw material is inserted in the vessel, followed by water injection as performed by Zhang and coworkers [83]. These authors pretreated corn zein with subcritical water in a batch reactor and observed positive improvements in the protein's solubility, thermal stability, and denaturation temperature.

According to Moraes and coworkers [84], extraction operated with supercritical technology at batch mode is not desirable in the industry because the pauses required between batches to remove the spent raw material and inserting a new material for processing increase operation times and reduce productivity.

### 5.1.3. Pressurized Liquid Extraction

Pressurized liquid extraction (PLE) is used to extract analytes from a solid or semi-solid matrix using compressed liquids below the subcritical state. Solvent mixtures containing water and ethanol were enough to intensify the extraction of bioactive compounds from underutilized vegetable products including phenolic compounds, and carotenoids.

Ethanol, water, and mixtures are the preferable solvents in PLE because they are generally considered as safe (GRAS). To the best of our knowledge, no recent studies on the PLE applied to yellow corn milling byproducts exist.

### 5.1.4. Chemical Reactions: Transesterification and Gasification

Transesterification or alcoholysis is the catalytic conversion of a mixture of triglycerides into fatty acid alkyl esters (FAAE), i.e., biodiesel. Supercritical $CO_2$ was shown to be a green alternative to assist biological (lipases) and non-biological catalysts in the rapid conversion of triglycerides in FAEE [85].

The supercritical state induces high diffusivity which accelerates the action of catalysts. In transesterification, SC-$CO_2$ is not consumed. After system depressurization, it is separated immediately from the reaction products and may be recycled [19].

Yadav and coworkers [22] studied the transesterification of corn oil, and methanol assisted with SC-$CO_2$ catalyzed with Nafion NR50. The highest conversion of triglycerides into fatty acid methyl esters was correlated with the highest oil-to-methanol molar ratio [22].

Gasification converts biomass into a mixture of gases. Supercritical water is used to gasify high-moisture biomass, promoting high gas yields and low residual chars and tars. Supercritical water gasification (SWG) includes multiple reactions, including oxidation, methanation, hydrolysis, and water gas shift [86].

The SWG of corn straw catalyzed by $K_2CO_3$ produced a high amount of hydrogen and zero carbon monoxide [23]. One reason hypothesized by the authors is the difficulty in the formation of carbon monoxide after lignin decomposition.

### 5.1.5. Micronization and Encapsulation

Supercritical technology is also used to improve the delivery of many active ingredients via micronization, encapsulation, and impregnation.

Micronization reduces the particle size of materials and improves solubility and the delivery of molecules. Encapsulation consists of incorporating the active ingredient with an encapsulating agent to enhance the shelf life of products. Multiple methods with supercritical fluids to precipitate particles via micronization and encapsulation include supercritical antisolvent fractionation (SAF), supercritical antisolvent (SAS), the rapid expansion of supercritical solutions, gas anti-solvent processes, and precipitation from gas-saturated solutions [87,88].

Rosa and coworkers [18] used zein as an encapsulant polymer to incorporate vitamin complexes after precipitation assisted with SC-$CO_2$ as antisolvent, generating microparticles with maximum incorporation of 13.74 mg riboflavin/g, 0.47 mg δ-tocopherol/g, and 14.57 mg β-carotene/g.

Palazzo and coworkers [89] used SC-$CO_2$ as co-solute to encapsulate the phenolic compound luteolin with zein incorporated with aqueous mixtures with ethanol and acetone, thus generating microparticles with a maximum encapsulation efficiency of 82%.

### 5.1.6. Impregnation and Extrusion

Corn starch is extensively used in research. Corn milling byproducts, including germ and the zein derived from gluten meal, are potential biopolymers for the formation of aerogels or alcogels to be dried with supercritical technology. In addition, formulations with such byproducts can be tested for printability in a 3D printer and subsequently incorporated with a drug of interest.

Liu and coworkers [90] studied the products generated after SC-$CO_2$-assisted impregnation of zein-based nanocomposites with cinnamon essential oil and suggested the products as long-term antibacterial delivery materials.

Dias and coworkers [91] incorporated beta-carotene in corn starch aerogels via SC-$CO_2$ impregnation (maximum yield of 0.96 mg beta-carotene/g aerogel) to enhance compound solubility in water. The poor solubility in water is related to the poor absorption of carotenoids in the intestine after consumption.

Extrusion of flour is highly performed to produce foods with distinct shapes and textures including snacks and pasta. Extrusion assisted with SC-$CO_2$ was also reviewed as a method to enhance the bioavailability of low-solubility drugs [92].

High-moisture corn fiber was extruded in a twin-screw extruder assisted with SC-$CO_2$. The total phenolic content in the products obtained in the extrusion assisted with $CO_2$ was statistically similar to the extrusion not assisted with SC-$CO_2$ for most of the conditions used [93]. The recently reviewed supercritical technology-based processes discussed in this review are summarized in Table 3.

**Table 3.** The supercritical-based processes used to valorize corn milling byproducts.

| Process | Byproduct | Conditions | Reference |
|---|---|---|---|
| SFE | Germ | Solvent: $CO_2$<br>Cosolvent: acetone, ethanol, and hexane<br>Temperature: 45–85 °C<br>Pressure: 15–25 MPa<br>$CO_2$ flow rate: 3 L/h<br>Cosolvent flow rate: 0.1 mL/min | [21] |
| SFE | Germ | Solvent: $CO_2$<br>Temperature: 80 °C<br>Pressure: 30 MPa<br>$CO_2$ flow rate: 2.5 L/min<br>Time: 10 min | [71] |
| SFE | Germ | Solvent: $CO_2$<br>Temperature: 45 °C<br>Pressure: 40 MPa<br>$CO_2$ flow rate: 5.6 g/min | [43] |
| SFE | Silk | Solvent: $CO_2$<br>Temperature: 40–60 °C<br>Pressure: 25–45 MPa<br>$CO_2$ flow rate: 20 L/h<br>Cosolvent flow rate: 1.3 mL/g<br>Time: 120 min | [79] |
| Hydrolysis assisted with SC-$CO_2$ | Stover: cob and stalk | Solvent: $CO_2$ and water present in raw material<br>Temperature: 40–70 °C<br>Pressure: 35–45 MPa<br>Time: Non identified | [80] |
| Hydrolysis assisted with SC-$CO_2$ | Stover: cob and stalk treated with ultrasound | Solvent: $CO_2$ and water present in raw material<br>Temperature: 170 °C<br>Pressure 20 MPa<br>Time: 30 min | [81] |
| Hydrolysis assisted with SW | Zein | Temperature:110–170 °C<br>Pressure: 0.1–0.8 MPa<br>Time: 20–120 min | [83] |
| Transesterification | Oil | Solvent: SC-$CO_2$<br>Temperature: 95 °C<br>Pressure: 9.65 MPa<br>Catalyst: Nafion NR50<br>Time: 240 min | [22] |
| SWG | Straw | Temperature: 450–550 °C<br>Pressure: 41 MPa<br>Catalyst: $K_2CO_3$<br>Time: 10–50 min | [23] |
| Encapsulation | Zein | Antisolvent: SC-$CO_2$<br>Solvent: Ethanol:water (94:6, *v/v*)<br>Temperature: 40 °C<br>Pressure: 7–16 MPa<br>Solution flow rate: 0.02 g/mL<br>$CO_2$ flow rate: 20–60 g/min | [18] |

**Table 3.** *Cont.*

| Process | Byproduct | Conditions | Reference |
|---|---|---|---|
| Encapsulation | Zein | Antisolvent: $CO_2$<br>Solvent: Aqueous mixtures with ethanol and acetone<br>Temperature: 60 °C<br>Pressure: 8.2–10 MPa<br>Solution flow rate: non-identified<br>$CO_2$ flow rate: non-identified | [89] |
| Impregnation | Zein | Temperature: 40 °C<br>Pressure: 15 MPa<br>Time: 60 min | [90] |
| Impregnation | Starch | Temperature: 40–60 °C<br>Pressure: 15–30 MPa<br>Number of cycles: 1–4<br>Depressurization rate: 0.25–2.61 MPa/min | [91] |
| Extrusion | Fiber | Temperature: 90–120 °C<br>Pressure: non-identified<br>Moisture: 30%<br>$CO_2$ flow rate: 200 mL/min<br>Screw speed: 100 g/min | [93] |

*5.2. Economic Evaluation*

To understand the feasibility of plant-based biorefinery, it is necessary (a) to know the composition of the feedstock, (b) to understand how the present compounds interconnect together for process design and optimization, and (c) to understand the costs involved in processes that are reflected in the economic value of products and (d) the environmental impacts provoked.

The cost of manufacturing (COM) is used to evaluate the economic feasibility of a process [94]. The COM consists of three major types of costs: (a) general expenses (GE): costs which cover business maintenance and include the costs involved in management, sales administration, and research and development; (b) fixed costs (FC): costs that do not depend on the production, i.e., land cost, insurance, territorial taxes, and depreciation; (c) operating costs (OC): costs that are dependent on the production and consist of raw material, operational labor, waste treatment, and utility costs.

The three components of COM are calculated in terms of five types of costs, i.e., the cost of raw material (CRM), the fixed capital investment (FCI), the cost of utilities (CUT), the cost of operational labor (COL), and the cost of waste treatment (CWT).

The major challenge in the scale-up of processes conducted with supercritical technology is the economic profitability of the process that is determined by the return on investment and the payback period which is the time to recover the cost of investment applied in the process. Based on the characteristics of bench-scale SFE of *Artemisia annua*, Baldino and coworkers [95] scaled up a 0.05 L bench SFE plant (equivalent to USD 21,195.00–29,673.00) to 50 L (equivalent to USD 317,925.00) by considering (a) operation at optimized conditions (residence time, amount of raw material, temperature, pressure, and $CO_2$ flow rate), (b) 30% costs for automation, as commonly used in the industry and (c) equipment plants reported in the literature.

For the SFE-assisted bypressing oil from Baru seeds, the total cost including equipment for a 0.1 L unit was calculated as USD 213,327.00. The analysis of the sensitivity of this process considering a 40 L plant with the cost of raw material as USD20/kg, the return on investment for the conditions studied was estimated to range between 15.15 and 937.44%, with an internal rate of return between 11.95 and 401.33%, and a payback time between 0.11 and 6 years [96].

Works on the process simulation and economic evaluation of the utilization of corn milling byproducts with supercritical technology are scarce.

Rosa and coworkers [18], who evaluated the encapsulation of vitamins with zein with SC-CO$_2$ as an antisolvent, observed that the quantity of CO$_2$ used for processing strongly affected the costs because of the electricity demand of operating the CO$_2$ pump. Considering the scenario of excess CO$_2$ utilization, the COM calculated for the vitamin microcapsules was USD 0.2/capsule; this value is competitive with the costs of commercially available multivitamin capsules. However, considering a reduction in 50% of CO$_2$ consumption for encapsulation, the COM could be reduced by approximately 24%.

Attard and coworkers [17] estimated the economic scenario for the extraction of waxes from corn stover with supercritical CO$_2$ based on the biorefinery approach. In the wax extracted with SFE, multiple unsaturated fatty acids were found. The COM estimated for wax was initially at USD 94.20/kg wax but decreased to USD 4.83/kg wax considering the scenario of the highest efficiency, i.e., to reuse biomass as combustion feedstock for energy generation [17].

Studies on the economic evaluation of using corn milling byproducts or other underutilized products should carefully evaluate aspects including the raw material, demand, that the market the products are to be obtained in is focused, the country, the quality requirements, and the local fees, among other issues, in order to provide a realistic scenario to companies and stakeholders [76].

### 5.3. Patents Survey

Patents are exclusive rights granted by a country to inventors that contribute to stopping others from taking advantage of their ideas without permission. The patented processes that valorize corn milling byproducts with supercritical technology include extraction, fractionation, and chemical reactions.

The SFE at bench and at industrial scale was patented to deodorize corn starch by removing "cardboard- or cereal-like" off-flavors associated with the compounds hexanal, 2-heptanone, trimethylbenzene, nonanal, and BHT-aldehyde [97].

The bench scale equipment consisted of a pump, extraction vessel, oven, CO$_2$ regulating valve, and an extract collection vial. The solid starch to be purified is tightly packed in the extraction vessel and placed in the oven. The oven is closed and heated to the extraction temperature, and pre-heated CO$_2$ is pumped into the vessel with the CO$_2$-flow regulating valve closed, until the target processing conditions are reached. For the industrial scale, the inventors inserted starch slurry (20–50% solids) into a countercurrent extractor but found limitations attributed to starch gelatinization. The inventors suggested using ethanol (as the cosolvent) to overcome this limitation in isolated starches and starchy flours.

Another process with SC-CO$_2$ was patented to reuse corn germ to enhance the yield of oil up to approximately 20% using SFE equipment consisting of at least one extractor vessel, one heat source, one pressure generator (pump), and indicators of temperature and pressure [98].

The patent CN101077990A integrated SFE of corn germ in an extractor vessel followed by fractionation of extract into two separators that generate different fractions of vitamins and unsaturated fatty acids. The equipment also contained a device to clean CO$_2$ after extraction and further recycle CO$_2$ in the storage tank to decrease processing costs [99].

A process patented for the hydrolysis of lignocellulosic feedstock used a semi-batch mode by inserting corn stover in a small packed reactor that was processed with a supercritical ethanol-CO$_2$ mixture [100]. In another patent, NaOH (1%, *v/v*) followed by SC-CO$_2$ was used to pretreat corn straw to increase the release of polysaccharides to be converted to xylose via enzymatic hydrolysis [101].

In contrast, the CN102210356B patent introduces an enzymatic reaction of corn germ oil assisted with SC-CO$_2$ to synthesize structured lipids with medium-chain fatty acids by up to 32% [102]. Table 4 shows the conditions used on the patents searched for this review.

**Table 4.** The patents reviewed the reuse of corn byproducts with supercritical technology.

| Material | Process | Conditions Used | Product | Patent Number | Location | Reference |
|---|---|---|---|---|---|---|
| Starch (solid and slurry at 20–50% solids) | Supercritical fluid extraction | Solvent: $CO_2$ and ethanol<br>Temperature: 50–120 °C<br>Pressure: >30 MPa<br>Solvent to the raw material ratio: 1–10 | Deodorized starch | US8216628B | United States | [97] |
| Germ meal | Supercritical fluid extraction | Solvent: $CO_2$<br>Temperature: 20–110 °C<br>Pressure: 11–64 MPa<br>Solvent to the raw material ratio: 2–30 | Oil | US8603328B2 | United States | [98] |
| Germ meal | Supercritical fluid extraction coupled with fractionation | Solvent: $CO_2$<br>Extractor: 40 °C and 8 MPa–20 MPa<br>Separator 1: 35 °C and 8 MPa–6 MPa<br>Separator 2: 35 °C and 6 MPa<br>Flow rate: 20 L/h | Fractionated oils | CN101077990A | China | [99] |
| Stover | Supercritical hydrolysis | Solvent: water<br>Temperature: 264 °C<br>Pressure: 7.58 MPa<br>Time: 20 min | Monosaccharides | US8282738B2 | United States | [100] |
| Straw | Supercritical hydrolysis | Solvent: $CO_2$ and water/C1–C5 alcohol<br>Temperature: 35–70 °C<br>Pressure: Non identified<br>Time: 60 min | Monosaccharides | CN112708647A | China | [101] |
| Germ oil | Enzymatic reaction | Enzymatic reaction<br>Temperature: 45–65 °C<br>Catalyst: Novozym 435 at 3–5%<br>Pressure: 8–13 MPa<br>Mixing speed: 120 r/min,<br>Reaction time: 18–28 h | Structured lipid rich in caprylin | CN102210356B | China | [102] |

## 6. Current Trends and Opportunities for Future Studies

Combining technologies is a trend for process intensification via the generation of multiple products with competitive costs. Based on the vast overview of possibilities offered by supercritical technology to process milling byproducts explored in this review, we propose a scenario to support further research on process optimization and feasibility.

In the first stage of this scenario, a storage tank that captures the $CO_2$ generated in the fermentation reservoirs in dry milling (Figure 4A) was included as a $CO_2$ reservoir for integrated equipment that operates extraction and particle formation integrated with $CO_2$ cleaning and recycling to reduce $CO_2$ emission to the atmosphere.

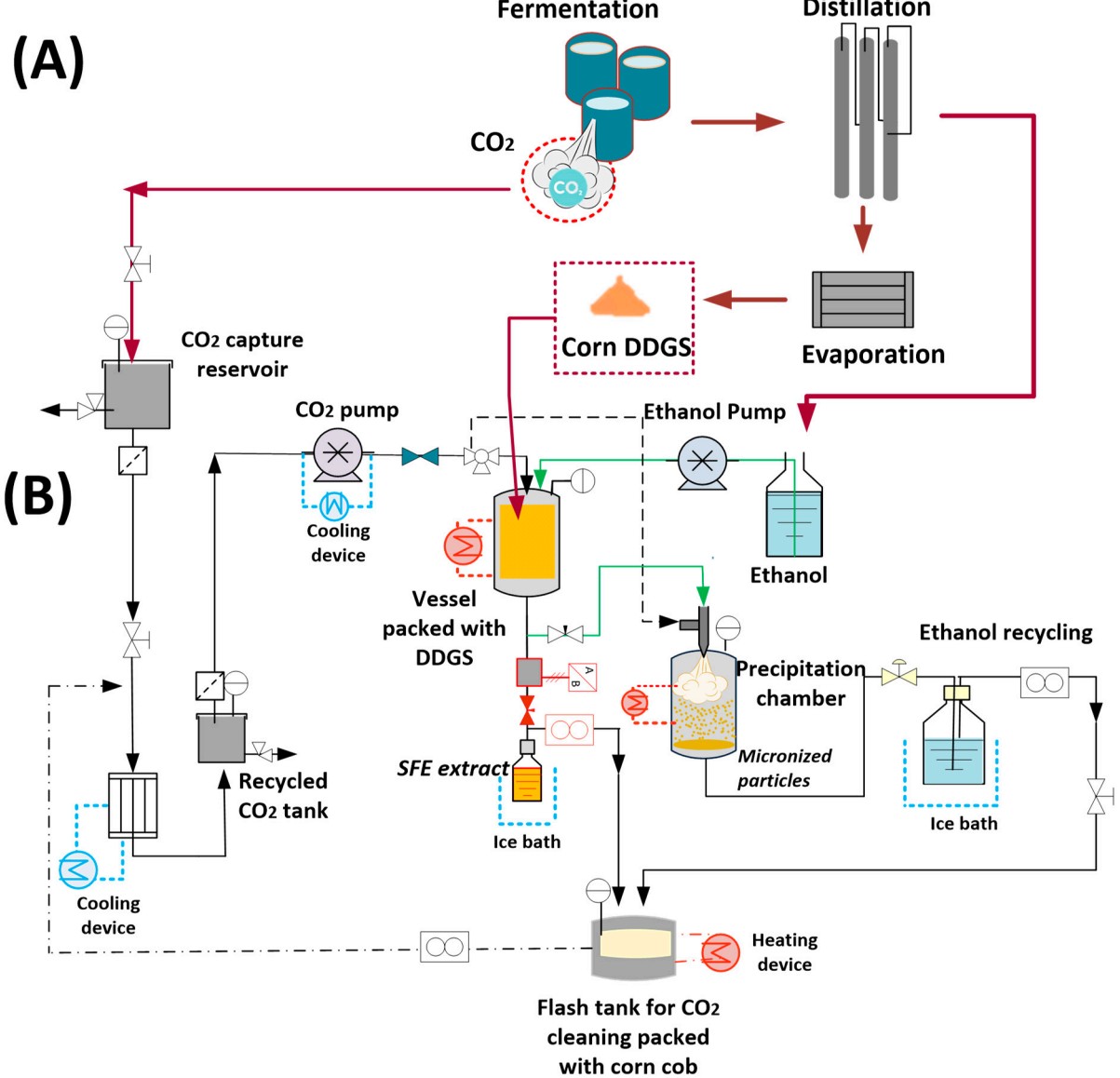

**Figure 4.** The future directions on the optimization of dry grinding (**A**) via integration with supercritical technology (**B**) for the reutilization of corn byproducts.

The $CO_2$ flows to the pump coupled to a cooling bath at $-5\,°C$ to maintain a liquid state. Liquid $CO_2$ is pumped to the heated extraction vessel packed with corn DDGS to reach the supercritical state. In this first stage, supercritical $CO_2$ flows to the extraction vessel to obtain an extract enriched with oil, carotenoids, phytosterols, and tocols.

In the second stage, the valve that allows the $CO_2$ flow is closed, and pressurized liquid extraction (PLE) starts with a stream of ethanol to select phenolic acids from semi-defatted corn DDGS (Figure 4B).

Afterward, the $CO_2$ valve is opened, $CO_2$ is pumped and interacts with the ethanolic extract that is transferred to a nozzle installed before the precipitation chamber that is used for micronization, i.e., particle formation via size reduction of extracts resulting in the concentration of target compounds.

Once all of the ethanolic extract is injected, only $CO_2$ is injected at constant flow to remove all traces of ethanol in the extract. Ethanol leaves the precipitation vessel and is recycled in a solvent reservoir, whereas the $CO_2$ is transferred to a flash tank packed with an adsorbent to remove all possible impurities carried by $CO_2$ and is subsequently coupled to a second storage tank with recycled $CO_2$. Figure 4 shows the schematic diagram of process intensification proposed in this review.

Torres and coworkers [16], who included a storage tank reservoir for SFE of annatto seeds, showed that the recycling of $CO_2$ reduces the cost of manufacturing from USD 349.50/kg to USD 122.67/kg via reduced $CO_2$ consumption by 85.24%. The semi-defatted and depigmented corn DDGS may be used as feedstock to synthesize bioactive peptides.

## 7. Conclusions

Milling transforms corn into multiple products and supports the biorefinery concept to valorize the underutilized fractions, including stover, condensed distillers' grains, and DDGS. In this work, we reviewed the opportunities offered by supercritical technology to valorize corn milling byproducts by discussing the patents deposited, the processes used and their economic feasibility, and the possibilities of being considered for future works.

For instance, the dry grinding integrated with supercritical technology was suggested in this review to valorize DDGS via the production of oil enriched with carotenoids, particles enriched with phenolic compounds, and a solid fraction to be reused as protein feedstock for the generation of peptides of high biological value.

The authors expect this review to support future research with milling optimization via integration with supercritical technology, by process optimization at bench scale, followed by economic evaluation.

**Author Contributions:** Conceptualization: Á.L.S. and M.A.A.M.; Methodology: Á.L.S.; Formal Analysis, Á.L.S.; Investigation: Á.L.S. and M.A.A.M.; Writing—Original Draft Preparation, Á.L.S.; Writing—review and editing: Á.L.S. and M.A.A.M.; Validation: Á.L.S. and M.A.A.M.; Visualization: Á.L.S. and M.A.A.M. All authors have read and agreed to the published version of the manuscript.

**Funding:** This research received no external funding.

**Data Availability Statement:** The data availability statement was included. Data analyzed in this review were a re-analysis of existing data, which are available at the works cited in the reference section.

**Conflicts of Interest:** The authors declare no conflict of interest.

## Nomenclature

| | |
|---|---|
| CDS | Condensed Distillers' Solubles |
| DDGS | Dried Distillers' Grains with Solubles |
| PLE | Pressurized liquid extraction |
| SC-$CO_2$ | Supercritical carbon dioxide |
| SFE | Supercritical Fluid Extraction |
| SWE | Subcritical Water Extraction |
| SWG | Supercritical Water Gasification |
| TS | Thin Stillage |
| WDG | Wet Distillers' Grains |

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
