# Peer review of "Valorization of Cereal Byproducts with Supercritical Technology: The Case of Corn"

_processes, doi:10.3390/pr11010289_

Round 1

Reviewer 1 Report

1. L11:  write down the full form of DDGS.

2. Novelty of the review should be included in the introduction.

3. All the abbreviated forms should be described during the 1st time used in the MS.

4. The extraction processes should be supported with proper diagrams, the following article may help the authors e.g. https://doi.org/10.1007/s12010-021-03631-8 

Author Response

ID Processes – 2159131 – Valorization of cereal byproducts with supercritical technology: the case of corn.

Dear Editor and Reviewers,

Authors are so grateful to the respected reviewers and editor for careful and constructive reading of our manuscript and providing valuable comments, which helped us to improve its quality. We carefully considered the comments and revised the manuscript accordingly. Detailed corrections have been separately provided point by point along with a clear indication within the revised text (in red color).

Reviewer #1

Comments and Suggestions for Authors

  1. L11:  write down the full form of DDGS.

As the reviewer suggested, the full form of DDGS acronym was inserted. Also, a list of nomenclatures was inserted at the end of the manuscript

  1. Novelty of the review should be included in the introduction.

The novelties were included in the abstract and on introduction.

  1. All the abbreviated forms should be described during the 1st time used in the MS.

Many thanks for the helpful comments. The revised manuscript followed the suggestion, i.e., the full form of acronyms written at first and further acronym used.

  1. The extraction processes should be supported with proper diagrams, the following article may help the authors e.g. https://doi.org/10.1007/s12010-021-03631-8

All the processes operated with supercritical technology mentioned in this review were organized in a table (Table 3 in the revised manuscript) for didactic purposes, according to the article recommended by the dear reviewer.

We hope that these corrections and modifications make the manuscript acceptable for the publication in Processes. We are surely ready to further revise it if other corrections are still required.

Sincerely yours

Dr. Ádina L Santana and Dr. M. Angela A. Meireles

Corresponding authors

Reviewer 2 Report

This review manuscript discusses the opportunities offered by supercritical technology to process corn milling byproducts, as well as the challenges to reach economic feasibility. Recently, supercritical technology valorized corn milling byproducts to obtain biodiesel, biogas, microcapsules, and extracts enriched nutrients, or to pretreat the solid matrix for further hydrolysis to produce monosaccharides and bioactive peptides. This review could be used as fundamental information for application of supercritical fluid (CO2) to extract bioactive compound or component from corn. However, some issues need to be addressed before acceptance as follows;

1.    Table 1: sum of all fractions in each material should be 100% or 100 g or not. For example, stover: moisture = 78.09, protein = 94.99, fiber = 35.46. Please check numbers in this table.

2.    Table 3: Please add product of each patent. 

Author Response

ID Processes – 2159131 – Valorization of cereal byproducts with supercritical technology: the case of corn.

Dear Editor and Reviewers,

Authors are so grateful to the respected reviewers and editor for careful and constructive reading of our manuscript and providing valuable comments, which helped us to improve its quality. We carefully considered the comments and revised the manuscript accordingly. Detailed corrections have been separately provided point by point along with a clear indication within the revised text (in red color).

Reviewer 2

Comments and Suggestions for Authors

This review manuscript discusses the opportunities offered by supercritical technology to process corn milling byproducts, as well as the challenges to reach economic feasibility. Recently, supercritical technology valorized corn milling byproducts to obtain biodiesel, biogas, microcapsules, and extracts enriched nutrients, or to pretreat the solid matrix for further hydrolysis to produce monosaccharides and bioactive peptides. This review could be used as fundamental information for application of supercritical fluid (CO2) to extract bioactive compound or component from corn. However, some issues need to be addressed before acceptance as follows.

Many thanks for your positive outlook and helpful comments.

  1. Table 1: sum of all fractions in each material should be 100% or 100 g or not. For example, stover: moisture = 78.09, protein = 94.99, fiber = 35.46. Please check numbers in this table.

Table 1 was reviewed and corrected accordingly. The sum of all revised fractions did not exceed 100% in the revised document. Another column for total and some carbohydrates found in corn products was also added.

  1. Table 3: Please add product of each patent.

As the reviewer suggested. Table 3 was revised accordingly. Please see Table 4 in the revised manuscript.

We hope that these corrections and modifications make the manuscript acceptable for the publication in Processes. We are surely ready to further revise it if other corrections are still required.

Sincerely yours

Dr. Ádina L Santana and Dr. M. Angela A. Meireles

Corresponding authors

Reviewer 3 Report

Interesting work, based on new and up-to-date scientific reports. It demonstrates an adequate understanding of the relevant literature in the field and cites an appropriate range of literature sources. Collecting information in tables is helpful in illustrating the topic. 

The presented tables and figures, which are very comprehensive, make a very good impression.

Please correct the type of the manuscript with a review instead of an article. Also, it is important to analyze the grammar structure and punctuation (e.g line 36).

It is important to state clearly the implications for research, practice, and society.

I kindly recommend the next paper to be consulted for the introduction section: https://doi.org/10.3390/foods11162454

Author Response

ID Processes – 2159131 – Valorization of cereal byproducts with supercritical technology: the case of corn.

Dear Editor and Reviewers,

Authors are so grateful to the respected reviewers and editor for careful and constructive reading of our manuscript and providing valuable comments, which helped us to improve its quality. We carefully considered the comments and revised the manuscript accordingly. Detailed corrections have been separately provided point by point along with a clear indication within the revised text (in red color).

Reviewer 3

Interesting work, based on new and up-to-date scientific reports. It demonstrates an adequate understanding of the relevant literature in the field and cites an appropriate range of literature sources. Collecting information in tables is helpful in illustrating the topic. 

The presented tables and figures, which are very comprehensive, make a very good impression.

Many thanks for your positive outlook and helpful comments.

Please correct the type of the manuscript with a review instead of an article.

The type of manuscript was already specified as Review at the moment of submission. The automatic conversion of manuscript in MS Word (.docx) to the mdpi template for peer-review provoked the mistake. The issue was already reported to the Assistant editor of the Special Issue.

Also, it is important to analyze the grammar structure and punctuation (e.g line 36).

Many thanks for the suggestion. All typos were corrected accordingly.

It is important to state clearly the implications for research, practice, and society.

In the introduction the importance of this review for research, practice and society was mentioned in the paragraphs 9-10 of the revised document.

I kindly recommend the next paper to be consulted for the introduction section: https://doi.org/10.3390/foods11162454

As the reviewer suggested, the manuscript was considered in our contextualization.

We hope that these corrections and modifications make the manuscript acceptable for the publication in Processes. We are surely ready to further revise it if other corrections are still required.

Sincerely yours

Dr. Ádina L Santana and Dr. M. Angela A. Meireles

Corresponding authors
